# CLOCK evolved in cnidaria to synchronize internal rhythms with diel environmental cues

**Raphael Aguillon[1,2]\*, Mieka Rinsky[1], Noa Simon-Blecher[1], Tirza Doniger[1], Lior Appelbaum[1,2], Oren Levy[1]\***

[1]Mina and Everard Goodman Faculty of Life Sciences, Bar-Ilan University, Ramat Gan, Israel; [2]The Multidisciplinary Brain Research Center, Bar-Ilan University, Ramat Gan, Israel

## Abstract

The circadian clock enables anticipation of the day/night cycle in animals ranging from cnidarians to mammals. Circadian rhythms are generated through a transcription-translation feedback loop (TTFL or pacemaker) with CLOCK as a conserved positive factor in animals. However, CLOCK's functional evolutionary origin and mechanism of action in basal animals are unknown. In the cnidarian *Nematostella vectensis*, pacemaker gene transcript levels, including *NvClk* (the *Clock* ortholog), appear arrhythmic under constant darkness, questioning the role of NvCLK. Utilizing CRISPR/Cas9, we generated a *NvClk* allele mutant (*NvClk*^Δ), revealing circadian behavior loss under constant dark (DD) or light (LL), while maintaining a 24 hr rhythm under light-dark condition (LD). Transcriptomics analysis revealed distinct rhythmic genes in wild-type (WT) polyps under LD compared to DD conditions. In LD, *NvClk*^Δ/Δ polyps exhibited comparable numbers of rhythmic genes, but were reduced in DD. Furthermore, under LD, the *NvClk*^Δ/Δ polyps showed alterations in temporal pacemaker gene expression, impacting their potential interactions. Additionally, differential expression of non-rhythmic genes associated with cell division and neuronal differentiation was observed. These findings revealed that a light-responsive pathway can partially compensate for circadian clock disruption, and that the *Clock* gene has evolved in cnidarians to synchronize rhythmic physiology and behavior with the diel rhythm of the earth's biosphere.

**\*For correspondence:**
raphael.aguillon@outlook.fr (RA);
oren.levy@biu.ac.il (OL)

**Competing interest:** The authors declare that no competing interests exist.

## eLife assessment

This **fundamental** study for the first time defines genetically the role of the *Clock* gene in basal metazoa, using the cnidarian *Nematostella vectensis*. With **convincing** evidence, the study provides insight into the early evolution of circadian clocks. *Clock* in this species is necessary for daily rhythms under constant conditions, but not under a rhythmic light/dark cycle, suggesting that the major role of the circadian oscillator in this species could be a stabilizing function under non-rhythmic environmental conditions.

## Introduction

Throughout the history of life on Earth, organisms have had to adapt to a constantly changing environment, including the ~24-hr daily rhythm of light/dark, driving the development of endogenous biological clocks. The circadian clock, entrained by external stimuli such as light, enables the organism to anticipate the onset of the light and dark phases and synchronize its physiology and behavior in harmony with the environment. This, in turn, enhances the organism's fitness and survival (*Hut and Beersma, 2011*; *Pittendrigh, 1993*; *Quiroga Artigas et al., 2018*). From single-celled organisms

to metazoans, circadian clocks have evolved multiple times, highlighting their importance to living organisms (*Hut and Beersma, 2011*; *Pittendrigh, 1993*). Despite the fundamental role circadian clocks play in regulating the rhythmicity of living organisms, their evolutionary origin and intricate molecular mechanisms remain ambiguousin early diverging animal lineages, such as cnidaria.

Rhythmic phenomena, including calcification, reproduction, and diel behavior patterns, have been examined in cnidarian species (*Quiroga Artigas et al., 2018*; *Taddei-Ferretti and Musio, 2000*; *Gutner-Hoch et al., 2016*; *Oren et al., 2015*; *Hoadley et al., 2011*; *Sorek et al., 2018*). While environmental stimuli, such as light, directly triggered some of these patterns, others persist in the absence of external cues, suggesting the presence of an internally generated and self-sustaining circadian clock (*Oren et al., 2015*; *Levy et al., 2007*; *Goffredo and Dubinsky, 2016*). At the molecular level, cnidarians possess homologs of putative core pacemaker genes found in bilaterians (*Reitzel et al., 2013*; *Reitzel et al., 2010*; *Shoguchi et al., 2013*). Several studies have shown that most of these genes display diel expression patterns under light/dark cycles. However, unlike most animals, their oscillation generally ceases without light cues (*Leach and Reitzel, 2019*; *Leach et al., 2018*; *Rinsky et al., 2022*) Thus, how the core pacemaker genes orchestrate rhythmic gene expression and circadian behaviors in cnidarians remains unclear.

One of the most studied cnidarian species in the field of chronobiology is the estuarine sea anemone, *Nematostella vectensis*. Few studies have shown that in diel lighting, the locomotor behavior of *Nematostella* has a ~24 hr rhythm that is maintained under constant conditions, suggesting it is regulated by an endogenous circadian clock (*Oren et al., 2015*; *Hendricks et al., 2012*; *Tarrant et al., 2019*; *Leach and Reitzel, 2020*). By this, the *Nematostella* genome codes for conserved core pacemaker genes such as *NvClk*, *NvCycle*, and the cryptochromes *NvCry1a* and *NvCry1b* (*Reitzel et al., 2010*; *Shoguchi et al., 2013*). The proposed circadian clock model in *Nematostella* is composed of the positive transcription factors (bHLH-PAS family), NvCLK and NvCYCLE, that heterodimerize and upregulate light-dependent cryptochrome genes in the feedback loop, and NvPAR-bZIPs in the feed-forward loop, which repress the transcription of the positive elements (*Reitzel et al., 2013*; *Reitzel et al., 2010*; *Leach and Reitzel, 2020*). More recently, the NvCLK-interacting pacemaker, NvCIPC, was predicted to act as an additional repressor of the NvCLK:NvCYCLE dimer (*Leach and Reitzel, 2020*). However, in contrast to the freerunning oscillation demonstrated for *Nematostella* behavior (*Oren et al., 2015*; *Hendricks et al., 2012*; *Leach and Reitzel, 2020*), transcriptional expression profiles of most candidate genes implicated in the pacemaker do not retain their oscillation period without light (*Leach and Reitzel, 2019*; *Leach et al., 2018*; *Leach and Reitzel, 2020*).

We employed the CRISPR/Cas9-mediated genome editing system to establish a *NvClk* mutant (*NvClk*$^{\Delta/\Delta}$) *Nematostella*. By combining behavioral monitoring and transcriptomic analysis, we aimed to elucidate the role of *NvClk* in regulating rhythmic locomotor activity and gene expression under varying light conditions. Our study revealed a robust light response pathway capable of compensation and a conserved function of CLOCK as a timekeeper without a light cue.

## Results
### Phylogenetic analysis and spatial expression pattern of *NvClk*

Phylogenetic analysis of NvCLK protein sequences positioned *NvClk* within the cnidarian branch (*Figure 1a*). It contains a basic helix–loop–helix (bHLH) DNA binding domain and two Per-Arnt-Sim (PAS) domains, similar to the protein structure found in other animals. PAS domains are crucial structural motifs in protein-protein interactions that drive the self-sustaining molecular mechanism underlying the circadian clock (*Hennig et al., 2009*; *Shearman et al., 2000*).

In situ hybridization chain reaction (HCRv.3) was performed to localize *NvClk* expression at the polyp stage. Polyps were sampled at ZT10, i.e., peak expression of *NvClk* (*Oren et al., 2015*; *Reitzel et al., 2010*; *Peres et al., 2014*). *NvClk* mRNA expression was observed throughout the animal tissue, and enriched expression was visible in the tentacle endodermis and mesenteries (*Figure 1b*). In contrast, no signal was observed in the negative control (*Figure 1—figure supplement 1a*). This expression pattern resembled the expression observed at the larvae stage (*Peres et al., 2014*). To date, functional manipulation of the *NvClk* gene has not been performed in basal animal lineages including, cnidarians, and its function is unknown in cnidarians (*Hennig et al., 2009*; *Shearman et al., 2000*; *Figure 1a*).

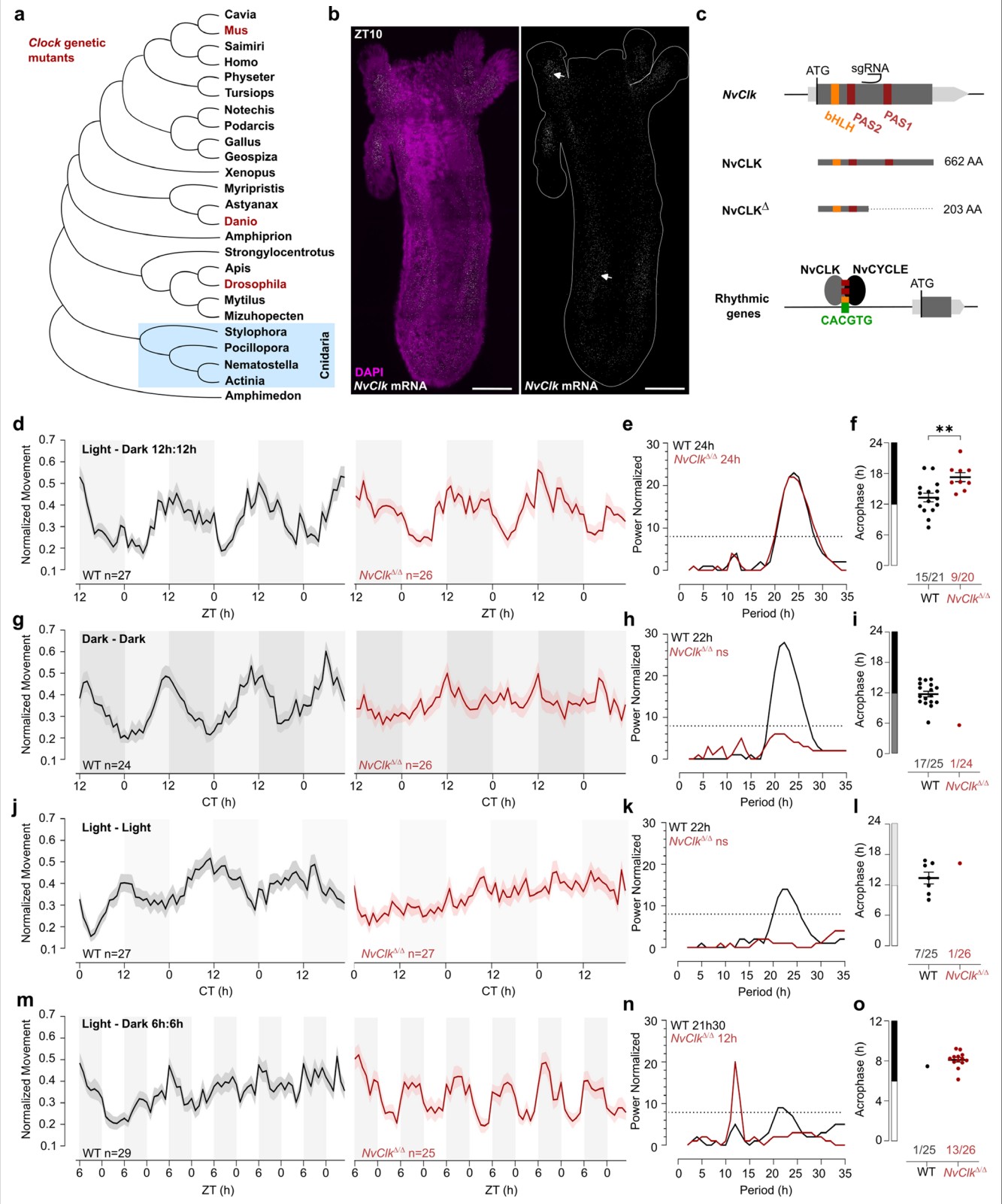

**Figure 1.** $NvClk^{\Delta/\Delta}$ cannot maintain circadian behavior in non-diel light conditions. (**a**) Phylogenetic tree showing the evolutionary relationship of CLK orthologs across different animal species. (**b**) In situ hybridization of $NvClk$ in the wild-type (WT) juvenile with scale bars representing 0.1 mm. (**c**) Schematic representation of the $NvClk$ gene in gray, with the open reading frame (ORF) in dark gray and the conserved protein domains basic helix–loop–helix (bHLH) (yellow) and PAS1 and PAS2 (dark red). The CRISPR-generated $NvClk^{\Delta}$ allele has a +20 nt insertion after the PAS1 domain, represented

*Figure 1 continued on next page*

*Figure 1 continued*

by a black arrowhead. NvCLK dimerizes via its three functional domains with NvCYCLE binding the CACGTG ebox to drive rhythmic transcription. (**d, g, j, m**). Normalized Movement (a.u), hourly binned over 72 hr, under different light conditions: 12 hr light:12 hr dark, continuous dark (Dark - Dark), continuous light (Light - Light), and 6 hr light:6 hr dark. The black line represents the WT, and the red line represents the *NvClk$^{\Delta/\Delta}$* mutant. (**e, h, k, n**) Lomb-Scargle Periodograms for each corresponding light condition. The significant period value (p<0.01) is indicated for each genotype in the top left corner of each graph. (**f, i, l, o**) Phase detection (Cosinor) and genotype comparison of 24 hr rhythmic individuals. See the number rhythmic/total on the x-axis indicating the number of 24 hr-rhythmic animals over the total population for each genotype.

The online version of this article includes the following figure supplement(s) for figure 1:

**Figure supplement 1.** NvClk$^{\Delta/\Delta}$ cannot maintain circadian behavior in non-diel light conditions.

## Generation of *NvClk$^{\Delta/\Delta}$* *Nematostella*

To investigate the function of *NvClk* in Nematostella, we employed the CRISPR-Cas9 system to generate mutants. Based on existing knowledge from mouse and *Drosophila* models, we hypothesized that NvCLK:NvCYCLE dimer binds to the DNA motif CACGTG within the promoter of rhythmic target genes (*Figure 1c*). Guide RNA (gRNA) was synthesized to target a region between the two PAS domains of the *NvClk* coding sequence (CDS). This gRNA and the Cas9 endonuclease were microinjected into zygotes (Methods). Subsequently, F0 animals were outcrossed with WT, and the F1 progeny were raised to adulthood. Genotyping of F1 polyps identified ten different mutated alleles, with six displaying a frame-shift mutation, including one with a 20 bp insertion (*NvClk$^{\Delta}$*), resulting in a premature stop codon. The predicted 203 amino acid truncated protein lacked 459 amino acids, including one co-factor dimerization PAS domain (*Figure 1c*, *Figure 1—figure supplement 1b*). To obtain homozygous *NvClk$^{\Delta/\Delta}$*polyps, we crossed heterozygous *NvClk$^{\Delta}$*F1 animals. Genotyping of F2 polyps confirmed the expected 25% frequency of *NvClk$^{\Delta/\Delta}$* mutants. Subsequently, we intercrossed *NvClk$^{\Delta/\Delta}$*animals to obtain F3 *NvClk$^{\Delta/\Delta}$* polyps for use in subsequent experiments aimed at assessing the role of *NvClk* in regulating behavioral and genetic rhythms.

## *NvClk* is necessary to maintain circadian behavior in constant conditions

To assess the impact of the *NvClk$^{\Delta}$* mutation on circadian rhythm, we monitored the locomotor behavior of WT and *NvClk$^{\Delta/\Delta}$* polyps under different light conditions (*Supplementary file 1*). Both the WT and *NvClk$^{\Delta/\Delta}$*populations exhibited a 24-hr periodicity in 12:12 hr LD cycles (*Figure 1d and e*), with 15 out of 21 WT animals displaying 24 hr rhythmicity compared to only 9 out of 20 *NvClk$^{\Delta/\Delta}$*animals (*Figure 1f*, *Table 1*, *Supplementary file 2*). The average acrophase for WT polyps (13.3 hr)

**Table 1.** Summary of rhythmic analysis of individual behavior.

| Cosinor | | **24 hr period** | **(p.adj <0.01)** |
|---|---|---|---|
| | WT | *NvClk$^{+/\Delta}$* | *NvClk$^{\Delta/\Delta}$* |
| LD | 15/21 | - | 9/20 |
| DD | 17/24 | 9/19 | 1/25 |
| LL | 7/25 | - | 1/26 |
| LD66 | 2/28 | - | 0/25 |
| DD66 | 1/27 | - | 0/24 |
| Cosinor | | **12 hr period** | **(p.adj <0.01)** |
| | WT | *NvClk$^{+/\Delta}$* | *NvClk$^{\Delta/\Delta}$* |
| LD | 1/21 | - | 0/20 |
| DD | 0/24 | 0/19 | 0/25 |
| LL | 0/25 | - | 0/26 |
| LD66 | 1/28 | - | 13/25 |
| DD66 | 0/27 | - | 0/24 |

was significantly earlier than for $NvClk^{\Delta/\Delta}$ polyps (17.3 hr) (*Figure 1f*, *Table 1*). While we could detect a 24-hr rhythm for both genotypes, the delayed acrophase and reduced number of significant rhythmic polyps in the $NvClk^{\Delta/\Delta}$ suggest an alterationof the underlying rhythmicity mechanism.

We then investigated locomotor behavior under continuous conditions, namely DD or LL. WT polyps exhibited a 22-hr rhythmic behavior under both constant light conditions, with 17 out of 25 WT polyps displaying a 24-hr rhythm under DD and 7 out of 25 under LL (*Figure 1g–l*). In contrast, a few $NvClk^{\Delta/\Delta}$polyps displayed rhythmic behavior under constant conditions (1 out of 24 in DD and 1 out of 26 in LL) (*Table 1*). Additionally, we observed an intermediate phenotype in the locomotor behavior of heterozygous polyps for the $NvClk^{\Delta}$ allele in DD (*Figure 1—figure supplement 1c–f*). These results revealed that $NvClk^{\Delta/\Delta}$ polyps could not maintain circadian rhythmicity without diel light cues.

A 24 hr rhythm of $NvClk^{\Delta/\Delta}$ polyps under LD conditions could be attributed to either direct light response or the partial functioning of the circadian clock due to the nature of the mutation. To distinguish between these two possibilities, we monitored locomotor activity under a 6-hr light: 6-hr dark (LD 6:6) cycle after a regular diel 72-hr entrainment under 12:12 LD. While WT polyps maintained a marginally significant periodicity of 22-hr, $NvClk^{\Delta/\Delta}$ polyps displayed a 12 hr rhythm at the population level (*Figure 1m–o*). Specifically, we identified a clear difference in 12-hr rhythmic individual polyps between WT and $NvClk^{\Delta/\Delta}$groups (1 out of 25 WT polyps vs. 13 out of 26 $NvClk^{\Delta/\Delta}$ polyps) (*Table 1*). Notably, entrainment with LD 6:6 did not lead to a 12-hr rhythm in DD for both WT and $NvClk^{\Delta/\Delta}$ polyps (*Figure 1—figure supplement 1g–i*). These results support the hypothesis that the 24-hr rhythm observed in the $NvClk^{\Delta/\Delta}$polyps in LD condition is due to the light-response pathway and not from an endogenous oscillator.

## *NvClk* regulates rhythmic gene expression differentially in response to light conditions

We conducted transcriptional profiling to investigate the underlying molecular correlates of the behavioral phenotype found in $NvClk^{\Delta/\Delta}$ polyps. WT and $NvClk^{\Delta/\Delta}$ polyps were sampled seven times every 4 hr over 24 hr under LD and DD conditions (*Figure 2a*). To identify rhythmic genes, we employed stringent statistical parameters, including Benjamini-Hochberg (BH.Q) for the JTK method (*Hughes et al., 2010*) and adjusted p-value (p.adj) for the RAIN method (*Thaben and Westermark, 2014*). This resulted in the identification of a minimal number of rhythmic genes. We detected only six rhythmic genes under LD conditions in WT polyps using the JTK method and 40 rhythmic genes using the RAIN method (*Supplementary file 3*).In DD condition, in the WT polyps, only two rhythmic genes were identified using the RAIN method (*Supplementary file 3*). Despite the risk of false positives, we opted not to use multiple testing but instead proposed to combine the JTK and RAIN algorithms to identify rhythmic genes, ensuring a robust approach to data analysis (p<0.01).We identified 119 rhythmic genes rhythmic under LD and 107 rhythmic genes under DD in WT polyps (*Figure 2b*, *Supplementary file 3*). In $NvClk^{\Delta/\Delta}$ polyps, we detected 147 rhythmic genes under LD and only 37 under DD (*Figure 2b*, *Supplementary file 3*).

The rhythmic genes in WT polyps displayed a delayed acrophase under DD compared to LD (17.20 hr vs. 12.93 hr, *Figure 2c*). However, no differences were detected between LD and DD rhythmic genes in $NvClk^{\Delta/\Delta}$ polyps (*Figure 2d*). Similarly, the relative amplitude (the gene amplitude divided by its baseline, rAMP) of DD rhythmic genes was higher in WT polyps compared to LD (0.61 vs 0.43, *Figure 2e*), but no rAMP difference was observed between LD and DD rhythmic genes in $NvClk^{\Delta/\Delta}$ polyps (*Figure 2f*).

Are rhythmic genes organized into 'transcriptional time clusters'? Does the $NvClk^{\Delta}$ mutation modify cluster recruitments, causing the loss of rhythmic behavior under DD conditions? We performed a clustering analysis on the rhythmic genes using the DPGP model (Dirichlet process Gaussian process mixture model). The number of genes per cluster between LD and DD conditions in WT polyps did not differ significantly (7.3 vs 7.6, *Figure 2—figure supplement 1*, *Supplementary file 4*). Interestingly, when clusters are organized by their acrophase, we observed clusters with higher numbers of genes peaking at subjective night in WT under DD conditions (*Figure 2—figure supplement 1*, *Supplementary file 4*). In $NvClk^{\Delta/\Delta}$ polyps, the number of genes per cluster was significantlyreduced in DD compared to the LD condition (4.1 vs 8.6, *Figure 2—figure supplement 1*). We did not identify GO-term enrichment in any cluster. However, the overlap between clusters and behavior opens new directions for further functional analysis (*Figure 2—figure supplement 2* and *Supplementary file 4*).

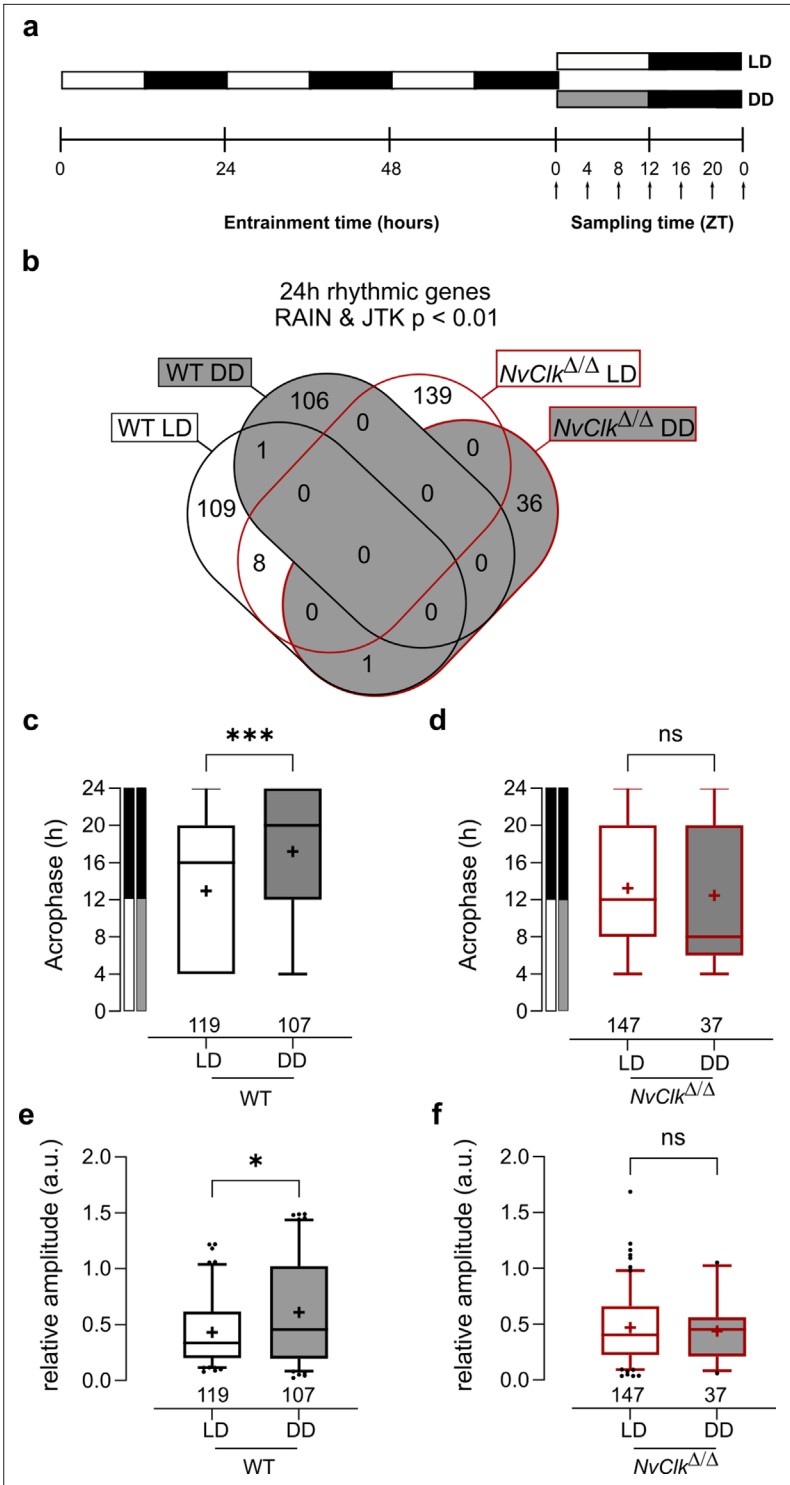

**Figure 2.** $NvClk^{\Delta/\Delta}$ shows rhythmic gene reduction in constant darkness with altered rhythmic features. (**a**) Overview of the experimental design used to generate RNA-seq data. Polyps were entrained for 72 hr before sampling at 4 hr intervals over a 24 hr period (dark arrows) in both light-dark (LD) and constant dark (DD) cycles. (**b**) Venn diagram comparing the total number of 24 hr rhythmic genes identified in wild-type (WT) and $NvClk^{\Delta/\Delta}$ in LD and DD cycles with a p<0.01 with RAIN and JTK. (**c**) Average acrophase comparison between rhythmic genes in LD and DD in WT polyps. Mann-Whitney test, p<0.001. (**d**) Average acrophase comparison between rhythmic genes in LD and DD in $NvClk^{\Delta/\Delta}$ polyps. Mann-Whitney test, p>0.05. (**e**) Average relative amplitude comparison between rhythmic genes in LD and DD in WT polyps. Mann-Whitney test, p<0.05. (**f**) Average relative amplitude

*Figure 2 continued on next page*

*Figure 2 continued*
comparison between rhythmic genes in LD and DD in *NvClk^{Δ/Δ}* polyps. Mann-Whitney test, p>0.05. (**c–f**) sample size (n) indicated below each boxplot.

The online version of this article includes the following figure supplement(s) for figure 2:

**Figure supplement 1.** Organization of transcriptional time clusters by phase relative to behavioral activity.

**Figure supplement 2.** Venn diagram showing overlaps of rhythmic genes in light-dark (LD) (wild-type, WT on the left and *NvClk^{Δ/Δ}* on the right) condition with rhythmic genes from *Leach and Reitzel, 2019* and the candidate pacemaker genes (based on protein conservation).

Overall, the reduced number of rhythmic genes in *NvClk^{Δ/Δ}* polyps under the DD condition and the reduced number of genesper cluster confirm the necessity of *NvClk* to recruit rhythmic genes in the DD condition and to organize them in transcriptional time clusters.

## *NvClk* regulates the temporal expression pattern of pacemaker genes

In line with previous findings in *Nematostella* (*Reitzel et al., 2010*; *Leach and Reitzel, 2019*), candidate pacemaker genes showed arrhythmic expression under DD conditions (*Figure 3a*, *Supplementary file 3*). However, the altered expression patterns observed in *NvClk^{Δ/Δ}* compared to WT polyps in LD conditionshowed increased transcripts for some genes (i.e. *NvClk* and *NvPar-bzipd*). In contrast, others (*NvCipc* and *NvPar-bzipc*) exhibited a reduction in transcript numbers (*Figure 3a*, *Supplementary file 3*). If we hypothesize that the first two genes (*NvClk* and *NvPar-bzipd*) act as positive factors and the latter two (*NvCipc* and *NvPar-bzipc*) potentially serve as negative regulator of the former, the lack of functionality of the *NvClk^{Δ}* allele would explain the observed difference in transcript levels between NvClk^{Δ/Δ} and WT polyps.

To systematically assess the mutation's impact on all the potential pacemaker genes, we utilized a correlation matrix based on their temporal transcript number levels, offering a comprehensive overview of their temporal organization. In WT polyps under LD conditions, the clustering categorized genes into twogroups: one exhibiting a daytime peaking, containing *NvClk*, and another peaking at night comprising *NvPar-bzipc* and *NvCipc*. Notably, in LD *NvClk^{Δ/Δ}* polyps, this second cluster contained two additional genes and displayed a weakened anticorrelation with the *NvClk* cluster (*Figure 3b*). These observations suggest that the pacemaker oscillation, generated by the interplay of positive and negative feedback loops, relies on the precise temporal organization of these potential pacemaker factors into distinct clusters. The disruption of this organization by the *NvClk^{Δ}* allele underscores the central role of *NvClk* in pacemaker function.

To go further into the regulatory mechanisms downstream of the pacemaker, we examined the presence of circadian E-box motifs (CACGTG) within 5 kb upstream of the predicted ATG of rhythmic genes. We calculated circadian/canonical E-box enrichment to account for the total variation in the number of canonical E-boxes (*Figure 3c*). Notably, only the candidate pacemaker genes exhibited a significant enrichment in circadian E-boxes in their promoters (15.9%) compared to the WT (5.6%), *NvClk^{Δ/Δ}* (4.8%) rhythmic genes, and non-rhythmic genes (6.8%) (*Figure 3d*).

## *NvClk* coordinates cell division and neuronal pathways in constant darkness

In addition to the transcriptomic rhythmic analysis, we aimed to identify processes regulated by *NvClk* that may not necessarily exhibit rhythmicity. We conducted a differential gene expression analysis on the total transcriptome between genotypes under each light condition to achieve this. Under LD conditions, *NvClk^{Δ/Δ}* polyps exhibited 457 down-regulated genes and 646 up-regulated genes, with no significant enrichment in GO terms observed (*Figure 4a*, *Supplementary file 4 and 5*). However, in DD conditions, *NvClk^{Δ/Δ}* displayed 2450 down-regulated genes and 1770 up-regulated genes (*Figure 4b*, *Supplementary file 4*). Notably, we identified enrichment in down-regulated genes in processes related to mitosis, microtubules, and ciliary/flagellar motility. Conversely, the up-regulated genes showed significant enrichment in processes such as the modulation of another organism's processes, axonal guidance, and sensory perception (*Figure 4b*, *Supplementary file 5*).

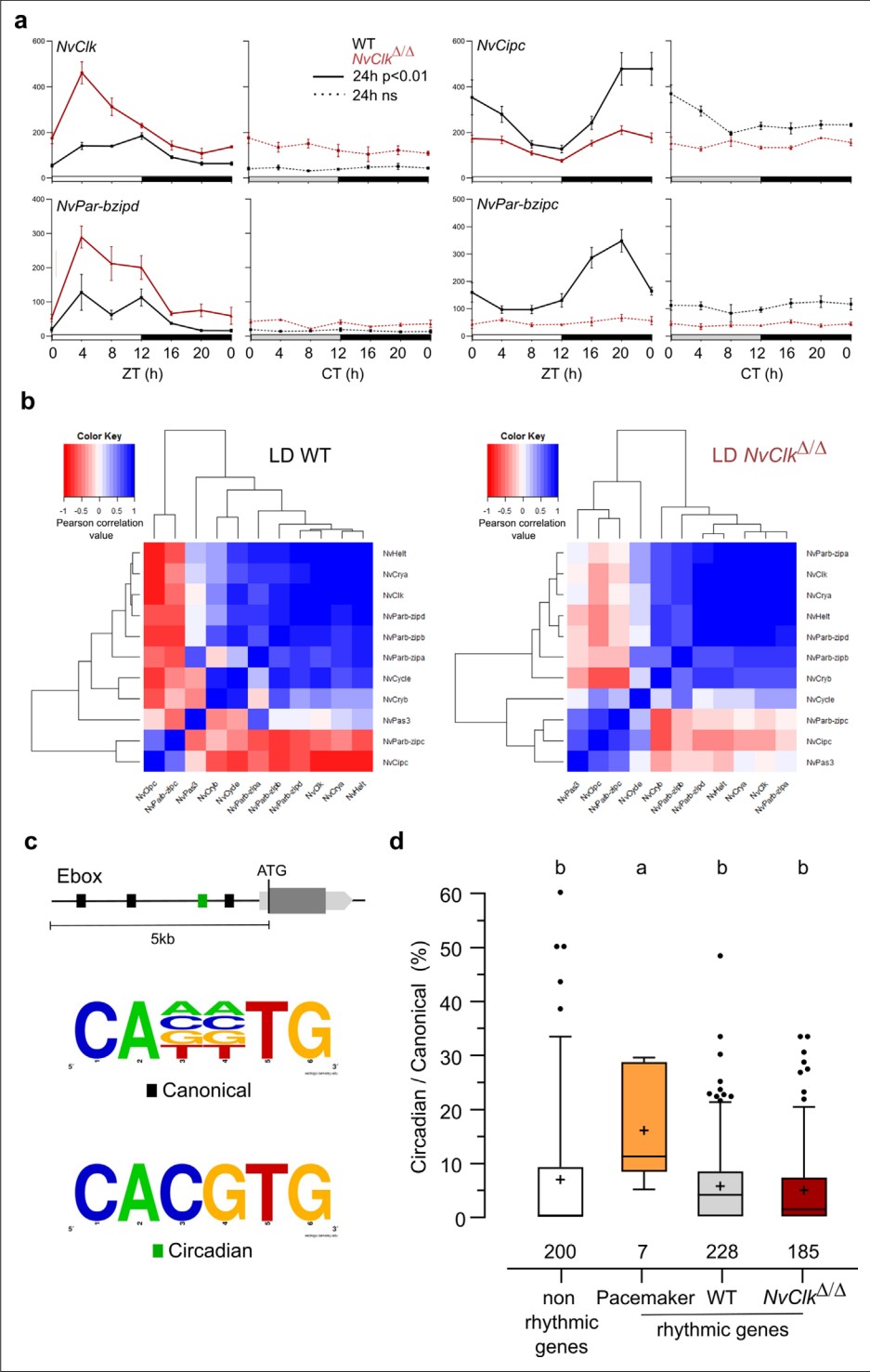

**Figure 3.** *NvClk^{Δ/Δ}* alters temporal pacemaker gene expression. (**a**) Four pacemaker genes are plotted, showing the read counts over 24 hr in light-dark (LD) and constant dark (DD) in wild-type (WT) (black) and *NvClk^{Δ/Δ}* (red). The continuous line represents significant rhythmicity (RAIN & JTK p<0.01), while the dashed line indicates no rhythmicity. (**b**) Correlation matrix of candidate pacemaker genes expression in LD for WT on the left and *NvClk^{Δ/Δ}* on the right. (**c**) Schematic representation of the promoter sequences analyses 5 kb upstream of the putative ATG. Black boxes represent canonical E-boxes, while circadian E-boxes are green. Below is the logo motif we used to identify canonical and circadian Ebox. (**d**) Circadian/Canonical ratio (in %) per condition. Kruskal-Wallis, multiple comparison, a vs b: p<0.05.

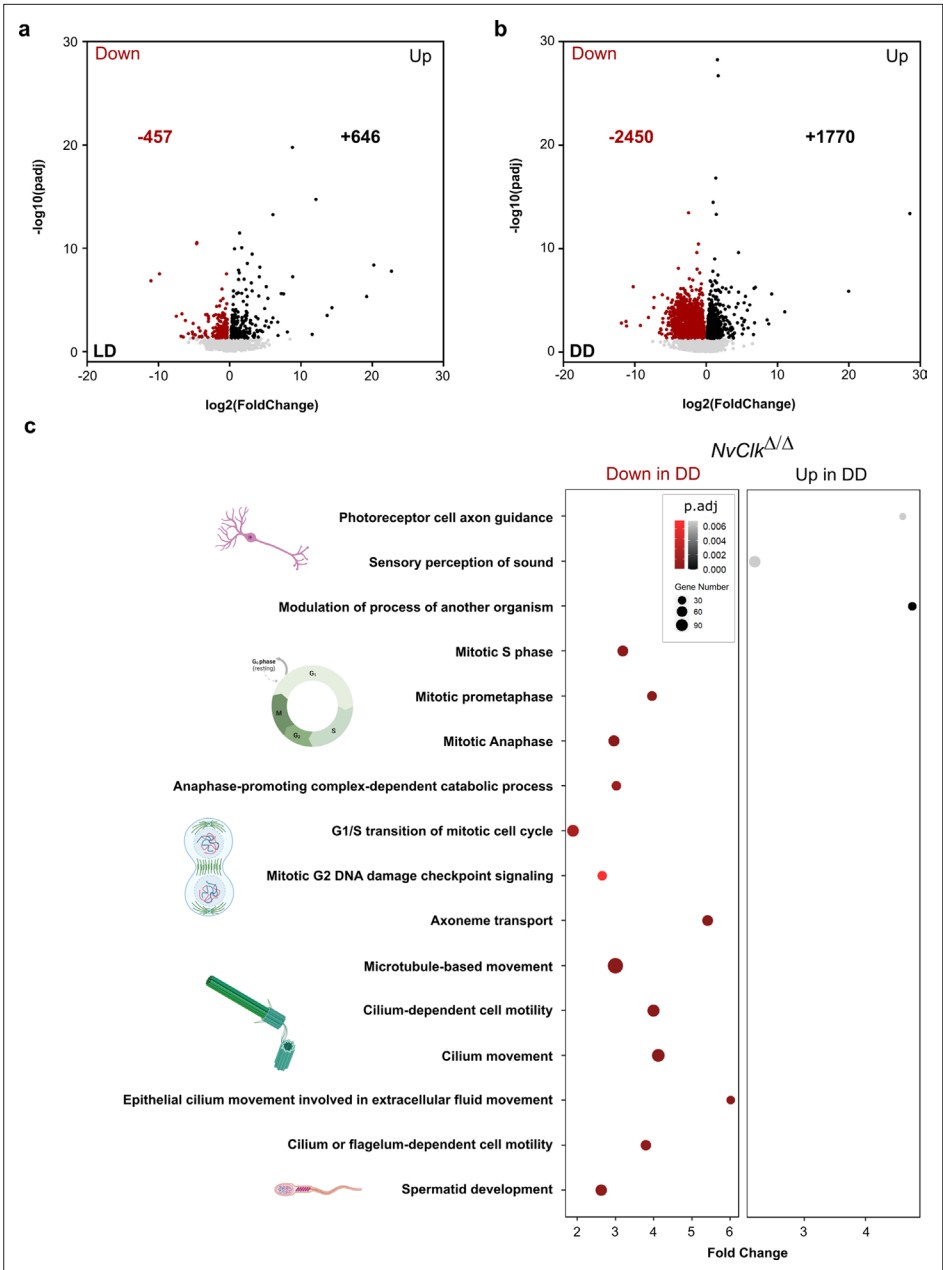

**Figure 4.** $NvClk^{\Delta/\Delta}$ disrupts cell-cycle and neuronal pathways in constant darkness. (**a, b**) Volcano plot showing the differential expression of genes (DEG) between wild-type (WT) and $NvClk^{\Delta/\Delta}$ in light-dark (LD) (left) and constant dark (DD) (right). Dashed line indicates the threshold used to detect DEG (p.adj<0.01). Red dots indicate down regulated genes and black dots up-regulated genes in $NvClk^{\Delta/\Delta}$ compare to WT polyps. (**c**) Gene Ontology (GO) terms with with significant fold-enrichment (Bonferroni corrected p-value or p.adjusted<0.01) for the DEG analysis in DD. Down regulated genes in Red while Up regulated genes in Black.

## Discussion
### Conserved behavioral CLOCK function through animal evolution

Our study provides valuable insights into the evolution of circadian clocks by characterizing the effects of the first *Clock* mutation in a cnidarian, the sea anemone *Nematostella vectensis*. Our behavioral assays showed that *NvClk* is essential for maintaining rhythmic locomotor activity without an entraining light cue. Although the rhythmicity of the $NvClk^{+/\Delta}$ heterozygote polyps was affected in DD, our results could not discriminate a dominant-negative from a total loss of function to identify the nature of this

mutation (*Figure 1—figure supplement 1g–i*). Studies in various model organisms further support the importance of CLOCK in regulating circadian locomotion. For instance, both *DmClk*<sup>Jrk/Jrk</sup>and *DmClk*<sup>ar/ar</sup>mutant flies exhibit a loss of circadian locomotion in constant darkness (*Allada et al., 2003*; *Allada et al., 1998*). Interestingly, the heterozygote for the allele *DmClk*<sup>Jrk</sup>, a dominant-negative mutation, had similar consequences on fly's behavior to our observation of *NvClk*<sup>Δ/+</sup>polyps behavior under DD conditions (*Allada et al., 1998*) suggesting that shortened CLOCK protein have the potential to be dominant-negative (*Figure 1—figure supplement 1g–i*). Within the vertebrate, then*DnClk1a*<sup>dg3/dg3</sup> zebrafish mutant displayed a shortened period under the same conditions (*Tan et al., 2008*). The dominant-negative mutant *MmClock*<sup>Δ5-6/Δ5-6</sup> mice showed a lossnof circadian locomotion in constant darkness, however, the complete deletion of the *MmClock* gene did not affect the circadian behavior rhythm in constant darkness suggesting compensation by a paralog (*Debruyne et al., 2006*; *Vitaterna et al., 1994*; *Asher and Schibler, 2006*). Overall, these findings support a conserved role of CLOCK in preserving circadian behavioral rhythms in the absence of light cues across the distant *Nematostella*, flies, zebrafish, and mice.

Moreover, the conservation of a 24 hr locomotion rhythm in LD of the *NvClk*<sup>Δ/Δ</sup> polyps with a delayed acrophase revealed a light-response pathway independent of the circadian circuit, consistent with observations in other animal models (*Allada et al., 2003*; *Allada et al., 1998*; *Debruyne*

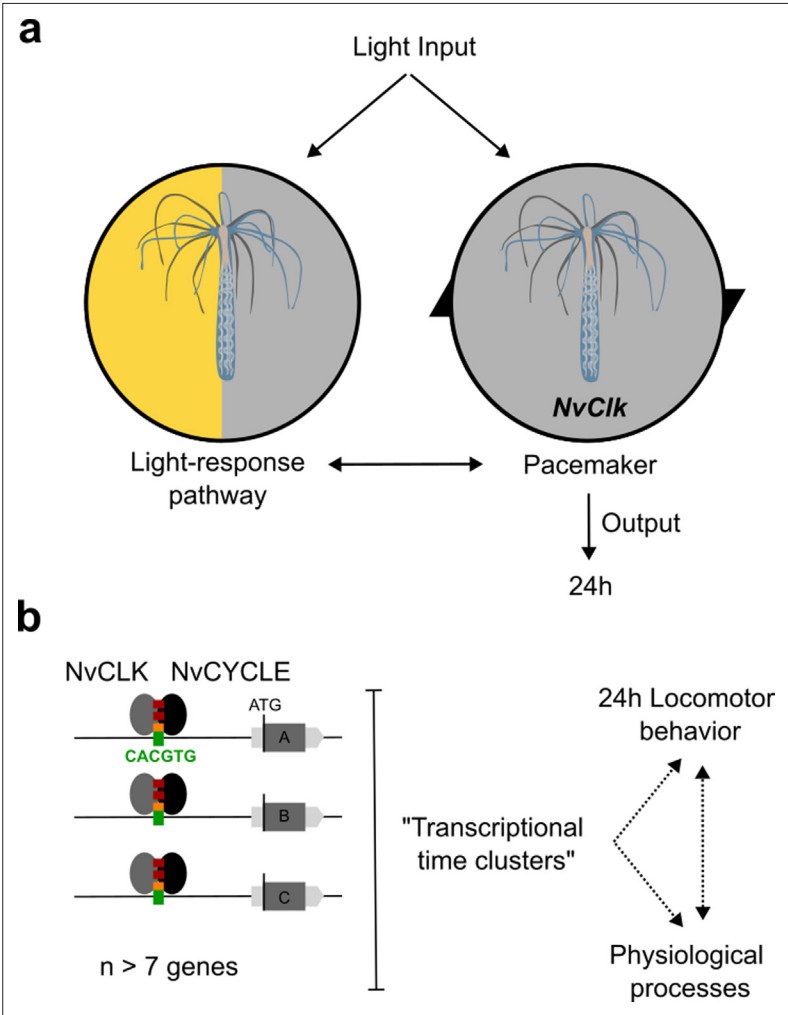

**Figure 5.** Summary of *NvClk* function in the regulation of *Nematostella* circadian rhythmicity. (**a**) The rhythmicity is the result of two interacting components, the pacemaker led by the *NvClk* gene and the Light-response pathway. In the wild-type (WT), the circadian clock overrides the Light-response pathway and imposes a circadian rhythm. (**b**) Downstream of the pacemaker, rhythmic genes are recruited by a group of seven genes in average. They might underly the rhythmic behavior and physiology of the polyp.

*et al., 2006*; *Figure 1f*). $NvClk^{\Delta/\Delta}$ polyps exposed to a 12:12 hr LD cycle exhibited a 24 hr period. In contrast, those exposed to a 6:6 hr LD cycle displayed a 12 hr period. Notably, nearly no WT polyps exhibited a 12 hr rhythm under this condition, suggesting that the circadian clock overrides the light-response pathway (*Figure 5a*). While some of the circadian factors can directly sense the light, such as CRY proteins (*Mat et al., 2024*), 29 putative *NvOpsin* have been identified in the genome which could be involved in the light-response pathway (*McCulloch et al., 2023*). Behavioral tracking of $NvClk^{\Delta/\Delta}$ polyps exposed to different wavelengths could help to identify candidates for further functional studies of the light-response pathway.

## Transcriptional rhythmicity plasticity downstream *NvClk*

At the transcriptomic level, previous studies in *Nematostella* have shown large changes in the transcriptional profile of many genes after a single day of constant darkness, including the candidate pacemaker genes that were found arrhythmic despite sustaining circadian locomotion (*Reitzel et al., 2010*; *Leach and Reitzel, 2019*; *Peres et al., 2014*). Consistent with previous transcriptomic analysis in cnidarian (*Hoadley et al., 2011*; *Leach and Reitzel, 2019*; *Rinsky et al., 2022*; *Brady et al., 2011*), most of the rhythmic genes identified in LD differed from those identified in DD in the WT polyps. Notably, they displayed higher mean acrophase and larger mean amplitude in DD, suggesting a differential regulation in response to light conditions, which had not been investigated in previous cnidarian studies. Furthermore, the overlap observed between our LD rhythmic genes and those identified by *Leach and Reitzel, 2019* underscores the robustness of pacemaker rhythmic transcription in LD conditions (*Figure 2—figure supplement 2*). However, the lack of overlap for rhythmic genes downstream of the pacemaker raises intriguing questions. Differences in experimental conditions, including genetic backgrounds, light system (Neon vs. LED), salinity (12ppt vs. 15ppt), and temperature (17 °C vs. 25 °C), may contribute to these discrepancies. Further investigations are necessary to determine if the lack of overlap of rhythmic genes downstream of the potential pacemaker genes results from an organism's adaptation to its environmentand, therefore, reflects the plasticity of the pacemaker in regulating its downstream rhythmic genes.

Our study identified 24 hr rhythmic behavior in $NvClk^{\Delta/\Delta}$ polyps under LD conditions, suggesting an alternative mechanism for generating molecular rhythmicity via the light-sensing pathway. However, it is crucial to note the minimal overlap between the rhythmic genes identified in $NvClk^{\Delta/\Delta}$ and WT polyps under LD conditions. This discrepancy indicates that the light-response pathway may not fully replicate the normal pacemaker functions observed in WT polyps, highlighting the need for further investigation into the recruitment and function of these genes. Additionally, the reduced number of rhythmic genes identified in $NvClk^{\Delta/\Delta}$ polyps under the DD condition underscores the crucial role of *NvClk* in maintaining molecular rhythm without light cues.

The clustering analysis revealed that rhythmic genes can be categorized into 'transcriptional time clusters'(aka synexpression clusters) *Rinsky et al., 2022*; *Niehrs and Pollet, 1999* by a group of seven/eight genes on average in the WT (*Figure 5b*). Their existence raises a fundamental question that has yet to be answered: How is a group of genes co-regulated in time and space (cell types) by the pacemaker? Their recruitment is disrupted in the DD $NvClk^{\Delta/\Delta}$ polyps suggesting an essential function of *NvClk* in the absence of light. The combination of published scAtlas (*Speir et al., 2021*) and multiplexed FISH techniques (*Choi et al., 2018*) will be essential to further investigate the biological regulation and function of these transcriptional time clusters.

## *NvClk* temporally organizes pacemaker gene expression to drive rhythmic gene recruitment

Our study reveals that *NvClk* plays a crucial role in regulating the temporal transcription of pacemaker candidate genes (*Figure 3a*). Our analysis identified two clusters of pacemaker genes: One containing *NvClk* and a second one containing a potential *NvClk* inhibitor (*NvCipc*) (*Zhao et al., 2007*; *Rivas et al., 2021*). These two clusters suggest the organization of the potential pacemaker genes transcription into interlock feedback loops with antiphase peaks, probably at the origin of the pacemaker oscillator function (*Shearman et al., 2000*; *Cyran et al., 2003*; *Dunlap, 1999*). The alteration of cluster composition with a weaker anticorrelation in LD $NvClk^{\Delta/\Delta}$ polyps might generate a desynchronization of the pacemaker factors' availability. Indeed, regulation of rhythmic transcription involved a complex protein-protein-DNA timing interaction. Furthermore, we did not identify any

circadian E-boxes enrichment in rhythmic genes between conditions, except for the candidate pacemaker genes. Altogether, this supports the function of *NvClk* in orchestrating the timing interaction of pacemaker factors to select downstream rhythmic genes, indicating a more complex regulatory landscape at play.

However, one significant unanswered question in our study is the reason for the arrhythmic transcription of putative pacemaker genes in DD. Using whole animals for sampling material might mask oscillating gene expression signals, especially if signals are present in a small number of cells or if tissues exhibit rhythmic gene expression in different phases. Furthermore, we must acknowledge a limitation in our interpretation, which is common in chronobiology: using RNA oscillation as a proxy for protein oscillation and function. The development of tools to study the pacemaker factors at the protein level in *Nematostella* will leverage this limitation in the field.

## *NvClk* regulates processes involved in cell proliferation and the neural system in the absence of light

Our study of *NvClk* suggests coordination of cellular processes, especially in the absence of light. Our rhythmic transcriptomic analysis results (*Figure 2* and *Figure 2—figure supplement 1*) raised questions regarding indirect effects and the non-rhythmic function of *NvClk*. We performed a differential gene expression analysis on the total transcriptome for each light condition. Under LD conditions, while*NvClk*$^{\Delta/\Delta}$ polyps exhibited significant changes in gene expression, we could not identify any GO term enrichment (*Figure 4a*, *Supplementary file 5*), revealing multiple altered processes we cannot yet identify.

In contrast, under DD conditions, *NvClk*$^{\Delta/\Delta}$ polyps displayed more pronounced alterations, with more DEGs and enriched GO-terms for down-regulated genes related to mitosis, microtubule organization, and ciliary/flagellar motility, while the up-regulated genes showed enrichment in processes such as the modulation of other organism's processes, axonal guidance, and sensory perception (*Figure 4b*, *Supplementary file 5*). These results imply that *NvClk* has non-circadian functions dependent on light availability. This is particularly noteworthy considering the expression of core pacemaker genes, known to be arrhythmic during larvae stages, potentially involved in developmental processes (*Peres et al., 2014*).

This study provides novel insights into circadian regulation in *Nematostella vectensis* and sheds light on the evolutionary origin of circadian time maintenance. Our findings indicate that CLOCK function is conserved from cnidaria to mammals to maintain rhythmicity without diel light cues. Furthermore, it revealed a light-response pathway able to compensate at both behavioral and molecular levels using light cues. This circadian clock mutant opens new avenues for investigating cell-type-specific mechanisms of the circadian clock that drive the molecular and phenotypical oscillations of cnidarians. By further exploring the circadian clock mechanisms in cnidarians, we can gain deeper insights into the evolutionary origins of this critical aspect of biology, enhancing our understanding of how organisms have evolved to keep track of time and adapt to their environment.

## Methods
### Animal husbandry
*Nematostella* were grown in 12 g.L sea salt water at 17 °C, and fed with Artemia *salina nauplii* three times a week. Spawning of gametes and fertilization was performed according to a published protocol (*Genikhovich and Technau, 2009*). In brief, the temperature was raised to 25 °C for 9 hr and the animals were exposed to strong white light. Three hours after the induction, oocytes were mixed with sperm to allow fertilization.

### CRISPR/Cas9 mediated mutagenesis
Genome editing in *Nematostella* was carried out following established CRISPR/Cas9 protocols, with slight modifications (*Hwang et al., 2013*; *Ikmi et al., 2014*). ZiFiT targeting software (http://zifit. partners.org/) (*Sander et al., 2010*) was used to select a guide RNA (gRNA) target site within the beginning of the *NvClk* exon and to design complementary oligonucleotides. To ensure the specificity of the gRNA, the selected target site sequence (GGTCCTCTCGTGGACTCTAC) was BLASTed against *Nematostella vectensis* genome (using JGI expected E value threshold of 0.1 to adjust for

short sequences: http://genome.jgi.doe.gov/Nemve1/Nemve1.home.html). To generate the gRNA template, the following oligonucleotides were used: Oligo 1: 5'- TAGGTCCTCTCGTGGACTCTAC –3' Oligo 2: 5'- AAACGTAGAGTCCACGAGAGGA –3' To construct the gRNA expression vector, pDR274 (plasmid # 42250; Addgene) was digested with the *BsaI* restriction enzyme. Subsequently, the gRNA oligonucleotides were annealed and cloned into the *BsaI*-digested pDR274 vector. Next, *DraI*-digested gRNA expression vectors, purified via ethanol precipitation followed by PureLink PCR purification kit (Invitrogen), were transcribed and purified using HiScribeT7 High Yield Transcription Kit (New England BioLabs) and illustra Microspin G-50 Columns (GE Healthcare Life Sciences), respectively. Cas9 recombinant protein with nuclear localization signal (260 ng/µl; PNA Bio, USA) was co-injected with the gRNA (140 ng/µl) into *Nematostella* zygotes. Injected embryos were raised in petri dishes at 22 °C under constant darkness with daily water changes.

## CRISPR/Cas9 mediated mutagenesis screening

To evaluate genome editing efficiency and mosaicism in F0 animals, genomic DNA flanking the guide sequence was amplified and Sanger sequenced. PCR was performed using two strategies. For the first, PCR reactions were performed using individual injected *Nematostella* (7 days post-fertilization), directly pipetted into a 25 µl PCR reaction containing a 2 x concentration of PCR MasterMix (Tiangen) and 10 pmol of each PCR primer. For the second, genomic DNA was extracted from tissue sampling from live adult animals, after relaxation in 7% $MgClphy_2$ (Sigma-Aldrich), using a NucleoSpin Tissue DNA purification kit (MACHERY-NAGEL). Subsequent PCR reactions were performed as above using 50 ng of genomic DNA. The primers used for these reactions (listed below) were designed to amplify a ~750 bp region around the targeted *NvClk* genomic locus. Mosaicism was determined if sequenced PCR products showed overlapping peaks in their chromatograms. The second strategy, which takes advantage of the ability of *Nematostella* to fully regenerate within a few days (*Reitzel et al., 2007*; *Burton and Finnerty, 2009*), is the one we refer to in the text hereafter. The injected individuals determined mosaic mutants were raised as F0 founders to sexual maturity and outcrossed with wild-type animals. The progeny of these crosses was raised and individually genotyped as described above. To determine inheritable mutations, sequences were further analyzed using the Tracking of Indels by DE composition web tool (TIDE). TIDE quantifies editing efficiency and identifies the predominant types of DNA insertions and deletions (indel) mutation composition from a heterogeneous PCR product compared to a wild-type sequence (*Brinkman et al., 2014*). Different heterozygous mutants were raised to sexual maturity and outcrossed with wild-type animals. The resulting F2 progenies were then raised to sexual maturity and genotyped before spawning for F3. Heterozygous mutants from each F2 progeny were intercrossed to obtain 25% homozygous F3 mutants. All animals used in this study are derived from heterozygous F2 mutants intercrosses, harboring the mutant allele $NvClk^{\Delta}$. PCR genotyping was performed using the following primer: Forward 5'- GATAAACACGGGCCGAAGATA –3' Reverse 5'- CAGTCCACGCTGGTCTAAAT –3'.

## Determination of $NvClk^{\Delta}$ F3 mutant genotypes

Genomic DNA was extracted as described above and used for following PCR and electrophoresis-based genotyping. PCR primers (listed below) encompassing the *NvClk* targeted site were used to produce PCR products of approximately 100 bp. The PCR products were then loaded and migrated by electrophoresis on a 3% Tris-borate-EDTA (TBE) agarose gel supplemented with GelStar Nucleic Acid Gel Stain (Lonza) for approximately 1 hr. The genotype of each F3 animal was determined by visualizing differences in migration speed of the PCR products caused by the CRISPR/Cas9 genome editing. The homozygous mutant animal ($NvClk^{\Delta/\Delta}$) produces only the larger ~120 bp amplicon while the wild-type animal ($NvClk^{+/+}$) produces only the lower ~100 bp amplicon. Animals heterozygous for the deletion ($NvClk^{+/\Delta}$) produce both the larger mutant and the smaller wild-type amplicons. PCR genotyping was confirmed by subsequent DNA sequencing of selected F3 animals. PCR was performed using the following primer: Forward 5'- ACCCCACTGAGTGACCTCTT –3' Reverse 5'- ATACGCCTGCGCTATACACC –3'.

## Behavioral assays

Locomotor activity of individual *Nematostella* were monitored using a lab-made setup equipped with an IP Infra-Red camera (Dahua Technology, Hangzhou, China), a white neon illumination (Aquastar

t8, Sylvania Lightning Solution) and constantly illuminated with low-intensity infrared (850 nm) LED light. The camera output 1 hr mp4 movie files which were AVI converted and then stitched. The data collection and analysis were carried out by EthoVision XT8 video tracking software (Noldus Information Technology, Wageningen, Netherlands). Animals were isolated in wells of six-well plates, each of which was manually defined as a tracking 'arena' in the EthoVision software. Center-point detection with gray scaling (detection range of 25–77, contour erosion of 1 pixel, high pixel smoothing) was used to monitor movements, which were calculated according to the change in position of the average center pixel each second. Illumination was provided with an intensity of 17 PPFD (+/−2) and did not significantly affect the experimental temperature (20°C). The illumination cycles were 12: 12 hr Light-Dark, 6: 6 hr Light-Dark or LL. Parameters were optimized to ensure that organisms were detected throughout the entire observation period.

### Behavior analysis

The total distance moved was summed in hourly bins and individually normalized min/max by the software GraphPad Prism 9.4. The average and standard errors were calculated for all tested animals based on the normalized values of each hour. The oscillation frequencies of the average population were evaluated based on the average values of each experiment using Fourier analysis-based software LSP with a $p<0.01$. For individual analysis we used the online platform Discorythm, combining different algorithms including Cosinor, JTK, and LSP (*Carlucci et al., 2019*). We chose Cosinor as it is the one designed to detect efficiently the acrophase.

### RNA-seq experimental design

All polyps were isolated in wells of six-well plates. Then, they were subjected to the 12: 12 hr LD cycle with 17 PPFD (+/−2) light intensity during 72 hr for entrainment in an incubator with a stable temperature at 18 °C. Subsequently, the polyps were divided into two experimental subgroups: 12: 12 hr LD and DD. Sampling began at 7 am (ZT0) and was performed at 4 hr intervals over 24 hr. At each time point, three or four individual polyps were sampled from each experimental group, immediately snap-frozen in liquid nitrogen, and transferred to −80 °C for storage.

### RNA extraction, library preparation, and sequencing

Total RNA was extracted from all sampled polyps (n=96) using TRIzol reagent (Invitrogen). Purified RNA samples were analyzed using a NanoDrop 1000 spectrophotometer (Thermo Fisher Scientific) to assess RNA quantity and 2200 TapeStation (Agilent) to assess RNA quality (RNA integrity number, >8.5). From each of the 96 samples four biological replicates in LD and three in DD, with the highest-quality extracts × 4 experimental subgroups × 7 time points, 1.5 μg of RNA was sent for library preparation (INCPM mRNAseq protocol) and sequencing at the Weizmann Institute Sequencing Unit, Israel. The libraries were sequenced using the bulk MARS-seq protocol (*Jaitin et al., 2014*, *Keren-Shaul et al., 2019*) on an Illumina NovaSeq 6000, resulting in an average of 17 million single-end reads of 113 bases per sample.

### Bioinformatic analysis

First, the unique molecular identifier (UMI) sequence of each read was extracted and placed within the read1 header file using UMI-tools extract (umi_tools v1.1). Next, the reads were mapped onto the *Nematostella* genome (NCBI genome GCA_000209225.1) using STAR (v2.6.0a) (*Dobin et al., 2013*) with default parameters. Mapped reads were then deduplicated based on UMIs using the umi_tools-dedup. The mapped reads were sorted by SAMtools (version 1.9). The number of reads per gene were quantified using HTSeq-Count (v0.12.4) (*Anders et al., 2015*).

### Rhythmicity analysis

Rhythmicity in transcript expression was assessed using the RAIN (ref-23) and metacycle (*Wu et al., 2016*) packages in R. The RAIN and JTK algorithms from metacyclewere run separately for each Nematostella genotype in both light conditions (LD and DD), treating them as individual datasets. All replicates (n=3) for each time point within a dataset were analyzed as regular time series to identify transcripts exhibiting daily oscillations. Specifically, we focused on transcripts with a precise 24 hr period, excluding those with a range (e.g. 10–14 or 20–28 hr). To improve the accuracy of identifying

true rhythmic genes, only transcripts with a p-value <0.01 in both RAIN and JTK analyses were deemed confidently cycling transcripts. Genes identified as significant cycling genes were subsequently utilized as input for the DPGP_cluster program (*McDowell et al., 2018*), which clusters genes based on their expression trajectories. Gene clusters comprising 10 or more genes underwent testing for GO term enrichment. Heatmaps were generated using the heatmap package (v4.5.5) in R. Venn diagrams were generated using the web tool Venn diagram (http://bioinformatics.psb.ugent.be/webtools/Venn/) And redraw with Inkscape. Expression plots were generated using GraphPad Prism (V.9.1).

## GO term enrichment analysis

After obtaining the differential gene expression results, Gene Ontology (GO) analysis was performed using the R TopGO package (v2.50.0). This analysis aimed to identify significantly enriched biological processes, cellular components, and molecular functions among the differentially expressed genes. The file 'nveGenes.vienna130208.GO_annotation_141017.txt' was utilized for GO analysis, and it was obtained from the following source: https://figshare.com/articles/dataset/Nematostella_vectensis_transcriptome_and_gene_models_v2_0/807696. This file contains the set of GO-transcript annotations that served as input for TopGO. The algorithm assigns a significance score to each GO term based on the enrichment p-value and the specificity of the term. In this study, the GO analysis was performed separately for the up-regulated and down-regulated genes in each condition (LD and DD) to identify the specific biological processes and molecular functions that are affected by the *NvClk*$^{\Delta}$ mutation.

## E-box motif enrichment analysis

Sequences for promoter regions (5000 kb upstream ATG) of differentially expressed genes were extracted. We manually identified in the list of motif enrichment all the E-box motifs and Circadian E-box motifs. Boxplots were generated using GraphPad Prism version 9.5.1.

## Differential expression analysis

Differential expression analysis was performed using R (v4.2.2) Bioconductor package, DESeq2 (v1.38.3) (*Love et al., 2014*). Raw read counts were obtained using HTSeq-Count (v0.12.4) (*Anders et al., 2015*) and then imported into DESeq2 for normalization and statistical analysis. Differentially expressed genes were identified using the Wald test with an adjusted p-value cutoff of 0.05. The analysis was performed on all the time points pooled of each genotype per light condition. The output of the analysis includes a list of genes with their log2 fold change, p-value, and adjusted p-value. Volcano plots were generated using GraphPad Prism version 9.5.1.

## HCR v.3 in situ hybridization

A custom *NvClk* (NVE2080, amplifier: B3 and B5) and *NvMyhc-st* probe set (NVE14552, amplifier: B5) were generated. We used *zfHcrt* probe set (ZDB-GENE-040324–1, amplifier: B1 and B3) as a negative control. For HCR on *Nematostella* juvenile, several alterations were made to a previously described protocol (*Choi et al., 2018*). Briefly, polyps were plucked and fixed in 4% PFA overnight at 4 °C. Polyps were washed 3x in 1x PBS and then dehydrated and permeabilized with 2×5 min washes in 100% methanol. The samples were stored at –20 °C overnight. To rehydrate the samples, a series of graded MeOH/PBST washes were used for 5 min each: 75% MeOH: 25% 1x PBS, 50% MeOH: 50% 1x PBST, 25% MeOH: 75% 1x PBST, and finally 2x washes in 100% 1x PBST. To further permeabilize the polyps, samples were incubated in 10 µg/ml Proteinase K diluted in 1x PBST for 10 min. Samples were quickly washed 3x in 1x PBST, and then post-fixed with 4% PFA for 10 min. After post-fixation, samples underwent 3×5 min washes with 1x 2x SSC + 0.1% Triton. From now, the following solutions (Pre-hybridization, hybridization, and probe wash buffers) were lab-made from the cnidarian-adapted hybridization buffer (*Sinigaglia et al., 2018*). Samples were then pre-hybridized with pre-hybridization buffer at 37 °C for 30 min. After pre-hybridization, samples were incubated with 2 pmol of the probe set diluted in hybridization buffer for 16 hr at 37 °C. To remove the probe mixture solution, samples were washed 2x for 30 min each with probe wash buffer at 37 °C. Samples were washed 2x for 5 min with 5x SSC + 0.1% Triton and then treated with probe amplification buffer for 30 min at room temperature. Samples were washed into a hairpin amplification buffer containing snap-cooled amplifier hairpins and were incubated at room temperature, protected from light, overnight. Samples were

then washed with successive 3x 5 x SSC + 0.1% Triton washes: 2x washes for 15 min. Nuclear staining was performed using DAPI 1:1000 in PBST for 1 hr. Samples were then washed with successive 2x 5 x SSC + 0.1% Triton washes: 2x washes for 5 min. Eventually were slide-mounted in glycerol and stored at 4 °C.

## Microscopy and image processing

Samples were imaged using a Zeiss LSM 710 with a 63x oil objective. They were slide-mounted in glycerol. Image manipulation was performed with Fiji (*Schindelin et al., 2012*). For the double probes *NvClk* imaging (*Figure 1b*), ROIs were generated from each *NvClk* probes signal and only the ROIs positive for the two fluorophores were kept. These ROIs were then used to extract from the original picture the signal considered as true mRNA signal. Figures were then assembled in Inkscape (http://www.inkscape.org/).

## Acknowledgements

We would like to thank the lab of Yehu Moran (The Hebrew University) for their help with advice on applying the CRISPR-Cas9 system in *Nematostella vectensis*. We thank Ms Roni Turgeman for her assistance with Nematostella cultures. We thank Dre Julie Batut for hosting RA during the manuscript revision of the manuscript and the Dre Elise Cau for her insightful comments. The research was funded by the Moore Foundation 'Unwinding the Circadian Clock in a Sea Anemone' (OL, grant no. 4598), and the Israel Science Foundation (LA, ISF, grant no. 961/19), we also acknowledge German Israeli Foundation GIF Nexus (OL, LA, no. G-1566-413.13/2023). Raphael Aguillon was funded by the Azrieli Foundation. This study represents partial fulfillment of the requirements for a Ph.D. thesis for M Rinsky at the Faculty of Life Sciences Bar-Ilan University, Israel.

## Additional information

### Funding

| Funder | Grant reference number | Author |
| --- | --- | --- |
| Azrieli Foundation | | Raphael Aguillon |
| Moore Family Foundation | 4598 | Oren Levy |
| Israel Science Foundation | 961/19 | Lior Appelbaum |
| German-Israeli Foundation for Scientific Research and Development | G-1566-413.13/2023 | Oren Levy Lior Appelbaum |

The funders had no role in study design, data collection and interpretation, or the decision to submit the work for publication.

### Author contributions

Raphael Aguillon, Mieka Rinsky, Conceptualization, Data curation, Formal analysis, Validation, Investigation, Visualization, Methodology, Writing – original draft, Writing – review and editing; Noa Simon-Blecher, Resources, Data curation, Formal analysis, Validation, Investigation, Visualization, Methodology; Tirza Doniger, Formal analysis, Investigation, Visualization; Lior Appelbaum, Funding acquisition, Validation, Writing – original draft; Oren Levy, Conceptualization, Resources, Supervision, Funding acquisition, Validation, Methodology, Writing – original draft, Project administration, Writing – review and editing

### Author ORCIDs

Raphael Aguillon ⓘ https://orcid.org/0000-0002-1149-0362
Lior Appelbaum ⓘ https://orcid.org/0000-0002-9248-5919
Oren Levy ⓘ https://orcid.org/0000-0002-5478-6307

Reviewer #2 (Public Review): https://doi.org/10.7554/eLife.89499.4.sa1

Author response https://doi.org/10.7554/eLife.89499.4.sa2

## Additional files

### Supplementary files
- MDAR checklist

- Supplementary file 1. This file contains the individual normalized locomotion behavior(normalizationon Min-Max values for each animals) for both genotypes in mutliple light condtions: WT, NvClk$^{D/+}$ (Htz), and NvClk$^{D/D}$ under LD 12: 12hr, DD, LL, LD 6: 6hr, DD 6: 6hr. Each experimental conditions were replicated independently three times.

- Supplementary file 2. This file contains the rhythmic analysis of individual polyp locomotor behavior of the different genotypes in different light conditions using https://mcarlucci. shinyapps.io/discorhythm/ using *Supplementary file 1* as input dataset. Only the Cosinor algorithm analysis is shown here. Animal with missing value were discarded by the software from the analysis which explain the discrepancy between total polyps tracks and individuals analyzed in some conditions.

- Supplementary file 3. This file contains the RNAseq rhythmic analysisbased on normalized read counts using the metar2 R package for rhythmic analysis for both genotypes under two light conditions: WT and NvClk$^{D/D}$ under LD 12: 12h and DD. The file containsthe list of potential core pacemaker candidate genes as well and if they were found rhythmic or not in each condition (p<0.01 RAIN and JTK).

- Supplementary file 4. This file contains the result of the clustering analysis of the rhythmic genes. Each cluster is identified by an ID number (from 1 to max), the associated ID NVE genes associated, their JGI annotation, and their acrophase.

- Supplementary file 5. This file contains the results of the Differential Expression analysis between WT and NvClk$^{D/D}$in LD, and WT and NvClk$^{D/D}$in DD. For each sample name w = WT and m = NvClk$^{D/D}$.

- Supplementary file 6. This file contains the results of GO term enrichment in the DEG identified in *Supplementary file 4*: WT and NvClk$^{D/D}$in LD, and WT and NvClk$^{D/D}$in DD. Color code: Blue is p.adj<0.01; Green is p.adj<0.05; Red is p.adj>0.05.

### Data availability
The RNA-seq data reported in this study have been deposited to the NCBI BioProject, under accession PRJNA935092. All data supporting the findings of this study are included in the manuscript and its supplementary files.

The following dataset was generated:

| Author(s) | Year | Dataset title | Dataset URL | Database and Identifier |
|---|---|---|---|---|
| Aguillon R, Rinsky M, Simon-Blecher N, Doniger T, Appelbaum L, Levy O | 2023 | Investigating the Function of the bHLH CLOCK in *Nematostella vectensis* | https://www.ncbi.nlm. nih.gov/bioproject/ PRJNA935092 | NCBI BioProject, PRJNA935092 |

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
