## [Editor Report · eLife assessment]

This **fundamental** study for the first time defines genetically the role of the *Clock* gene in basal metazoa, using the cnidarian *Nematostella vectensis*. With **convincing** evidence, the study provides insight into the early evolution of circadian clocks. *Clock* in this species is necessary for daily rhythms under constant conditions, but not under a rhythmic light/dark cycle, suggesting that the major role of the circadian oscillator in this species could be a stabilizing function under non-rhythmic environmental conditions.

---

## [Referee Report · Reviewer #2 (Public Review)]

In this revised manuscript Aguillon and collaborators convincingly demonstrating that CLK is required for free-running behavioral rhythms under constant conditions in the Cnidarian Nematostella. The results also convincingly show that CLK impacts rhythmic gene expression in this organism. This original work thus demonstrates that CLK was recruited very early during animal evolution in the circadian clock mechanism to optimize behavior and gene expression with the time-of-day.

---

## [Author Response]

The following is the authors’ response to the previous reviews.

**Public Reviews:**

**Reviewer #2 (Public Review):**
In this revised manuscript Aguillon and collaborators convincingly demonstrating that CLK is required for free-running behavioral rhythms under constant conditions in the Cnidarian Nematostella. The results also convincingly show that CLK impacts rhythmic gene expression in this organism. This original work thus demonstrates that CLK was recruited very early during animal evolution in the circadian clock mechanism to optimize behavior and gene expression with the time-of-day. The manuscript could still benefit from some improvements so that it is more accessible for a wide readership.
**Recommendations for the authors:**

**Reviewer #2 (Recommendations For The Authors):**
Aguillon and collaborators have deeply revised, and in the progress significantly improved the presentation of their interesting results with the first Cnidarian circadian gene mutant. Results are now very convincingly demonstrating that CLK is required for free-running behavioral rhythms under constant conditions. The results also now more convincingly show that CLK impact rhythmic gene expression, although interpretation of the transcriptomics data is not straightforward. I think there is still improvements that are needed to make the manuscript more accessible. We authors need to keep in mind that a broad audience will read their report, not just chronobiologists. I have listed below several issues that I think should be addressed, and some editing suggestions.

General comment to Editor and Reviewers:

We are genuinely grateful to both reviewers and editors about all the feedback which helped us to make the best of our data, to question our analysis to the point we redefined our approach and end up with a great article we are proud of it. Only the name of authors is visible on the article, and considering how much the reviewing system help to improve the research it seems almost unfair. As such, we thank all of you and really appreciate the new eLife system. Bravo all.

Abstract:(1) Line 40" It should read "transcript levels" instead of "transcription". There is no measurement of transcription rates in this manuscript, only mRNA levels.

Modified accordingly.

(2) Line 41: the authors mention "constant light". Does this refer to previous work? Their data in Figure 4 were in constant darkness, not in LL.

Modified accordingly.

(3) Line 46 and throughout the manuscript, the allelic nomenclature is not standard. 1-/- seems to indicate there are two different alleles. Since the allele might not be a null, I would suggest simply using 1/1, or perhaps delta/delta since the mutation results in a truncates CLK.

NvClk1-/- became NvClkΔ/Δ. Except in the .xls supplementary table were the mutant kept the NvClk-/- nomenclature. It is not possible to replace only part of a word with a different font, here generating delta sign would require to do it one by one.

(4) The last sentence of the abstract needs to be rephrased, as it suggests that CLK evolved to maintain circadian rhythms under constant conditions. Constant conditions very rarely exist on Earth, and thus cannot be an evolutionary driving force. Different explanations have been proposed on why a self-sustained clock is the evolutionary solution to timekeeping, but the purpose of the clock and of clock genes is not to maintain oscillations in constant conditions. Actually, this sentence conflicts with the title.

Modified to: the Clock gene has evolved in cnidarians to sustain 24-hour rhythmic physiology and behavior in absence of diel environmental conditions.From my actual understanding, you are right, the purpose of clock gene is not to maintain oscillation in constant conditions (this is simply the result of the experiment), but to synchronize the physiology to the day/night rhythm, and surely to sustain 24h oscillations in case the environment challenges the perception of the diel cues. The DD or LL is just an artificial experimental design to reveal the endogenous time-keeping pacemaker.

Results:(1) Line 148 and elsewhere in the MS: I would not use the word "lower" or "higher" to qualify acrophases. I would suggest advanced/delayed or earlier/later.

Modified accordingly.

(2) Line 157-9: The introductory sentence does not clearly present the rationale for the 6/6 experiments.

We modified the paragraph accordingly: The presence of a 24-hour rhythm of NvClkΔ/Δ polyps under LD conditions could be attributed to either a direct light-response or the partial functioning of the circadian clock due to the nature of the mutation….

(3) At the end of the behavior section, or perhaps at the end of each paragraph in this section, it would be helpful to have a summary of the results and more clearly explain their interpretation. The authors need to guide the readers, particularly non-chronobiologist, so that they can understand what the really neat data that were obtained mean. For example, what does it mean that the acrophase is different between mutant and wild-type, why are Clk mutants rhythmic under LD12/12 or 6/6, etc.

We added a conclusion sentence to help non-specialist to understand each result.

(4) Line 172 and elsewhere" "true rhythmic genes" sounds odd to me. Either they are, or they are not rhythmic.

Modified to “rhythmic genes.”

(5) Paragraph starting with line 184: I do not follow what is important about the number of genes per time cluster. What does it tell us, beyond the simple fact that less genes are rhythmic in the Clk mutants?

We rewrote the result paragraph to make it clearer why we performed this clustering analysis. This clustering analysis became Extended Data Fig.2 with modification of the figures (see my comments in your review about Figure 3).

(6) Line 197: The authors need to explain what they saw with circadian clock genes and their expression in CLk mutants. In some case, amplitude increased in LD. This surprisingobservation deserves some explanations. "Complex regulatory effect" is too vague.

We replaced the vague “complex regulatory effect” by a more thorough description of the figure 3.a.

(7) Line 198-203: Again, help the reader understand the significance of these observations.

We rewrote the paragraph to help the reader to better understand the significance of these observations.

Discussion:(1) Line 236-40. Careful with the use of -/-, which implies that an allele is a null. The first CLk mutants in mammals and flies, which the authors refer to. were actually dominant negatives.

I went over the citations we used for this paragraph and this first mutation in fly dClkar is null, no dominant negative. Flies are still rhythmic in the dark. Unless there is an older mutation? However, you right the first mutation identified in mouse was a dominant-negative with loss of rhythmicity, while the gene deletion did not show any effect on the behavior, suggesting compensation by a paralog. I removed two references which were not relevant to the discussion.

(2) Line 265-268 are not very clear. Do the authors mean that the lack of overlap for non-cricadian pacemaker genes is because of different experimental conditions? What would be those differences? It is reassuring that the Leach/Reitzel study and the present share pacemaker genes as rhythmic, but it is also surprising that there is almost no overlap beyond these genes. How robust are those other rhythms compared to circadian clock genes?

We revised this paragraph and raised major points regarding the raising condition of our polyps between labs and their potential genetic differences which could explain these differences.

(3) Line 270. I am not sure "compensation" is the right word, since there is no overlap between the rhythmic genes in mutants under LD and wild-type under either LD or DD. Also, saying on line 273 that the transcriptional pattern is not fully reproduced is a rather striking understatement, given the absence of rhythm gene overlap

We rewrote the paragraph accordingly. We replaced by “alternative way to drive rhythmicity under LD condition”.

(4) Line 279. The authors mention the possibility of false positives. Based on the FDR, is there more rhythmic genes than by chance?

The possibility of false-positive is a risk to consider when you do not perform multiple-testing. We added within the results paragraph the number of rhythmic genes identified with BH.Q or p.adj. which both are the multiple testing for each algorithm (RAIN and JTK) we used.

(5) Line 279-82. The references to the Ray study is rather obscure. What is the point the authors are trying to make here?

Eventually, we removed the reference from this article and modify the paragraph of the discussion. Indeed, the discussion around the Ray study did not gave an interesting direction to discuss our results and analysis approach.

(6) Line 284: define BHQ and p.adj

Defined and referenced.

(7) The way Lines 283-288 are worded do not provide a good rationale for how transcriptional rhythms were analyzed. The idea to combine two different approaches (JTK and RAIN) to be selective with rhythmicity was great. The authors need however to justify these choices in a more convincing manner. The goal is to detect rhythmic genes in a reliable manner, irrespective of the number of rhythmic genes observed Also, explaining the choice of methodology belongs to the result section.

We explained our choice of methodology and moved it to the result section as suggested.

(8) Line 292-3. There are known mechanisms that explain how transcriptional time clusters are generated. In particular, the use of interlocked feedback loop with antiphase peaks of transcriptions is well documented. Actually, it seems to me the clustering shown in Fig 4 might hint at such a mechanism.

Indeed you are right the clustering shown in Fig 3 (former Fig 4) revealed such mechanism.

Figures:Figure 2: Define relative amplitude

We added the definition of the relative amplitude within the results. If this is what you asked for?

Figure 3: Some of the cycles look odd (first row of graphs in panel C). Why would the first and last data point be so different in three of these graphs?

We decided to modify this figure as we realized it was not informative and not objective enough, as we selected among multiple patterns few “representatives”. In the new figure we combined the cluster analysis to the behavior. Thus, readers can now pick a cluster according to a specific behavior activity level (or ZT/CT) and reach in supp. Table 4 the “genes of potential interest”. However generally speaking this figure does not explain more the consequences of the mutation, so we moved it into the Extended data Fig.2

Figure4: define the color coding in the correlation panels (blue to red)

These values from -1 to 1 are the Pearson correlation values. Now indicated on the figure with the color coding legend.